# Remarkable flexibility in freestanding single-crystalline antiferroelectric PbZrO₃ membranes

Yunting Guo[1,4], Bin Peng [1,4] ✉, Guangming Lu[2,3,4], Guohua Dong[1], Guannan Yang[1], Bohan Chen[1], Ruibin Qiu[1], Haixia Liu[1], Butong Zhang[1], Yufei Yao[1], Yanan Zhao[1], Suzhi Li[3] ✉, Xiangdong Ding[3], Jun Sun[3] & Ming Liu[1] ✉

The ultrahigh flexibility and elasticity achieved in freestanding single-crystalline ferroelectric oxide membranes have attracted much attention recently. However, for antiferroelectric oxides, the flexibility limit and fundamental mechanism in their freestanding membranes are still not explored clearly. Here, we successfully fabricate freestanding single-crystalline PbZrO₃ membranes by a water-soluble sacrificial layer technique. They exhibit good antiferroelectricity and have a commensurate/incommensurate modulated microstructure. Moreover, they also have good shape recoverability when bending with a small radius of curvature (about 2.4 μm for the thickness of 120 nm), corresponding to a bending strain of 2.5%. They could tolerate a maximum bending strain as large as 3.5%, far beyond their bulk counterpart. Our atomistic simulations reveal that this remarkable flexibility originates from the antiferroelectric-ferroelectric phase transition with the aid of polarization rotation. This study not only suggests the mechanism of antiferroelectric oxides to achieve high flexibility but also paves the way for potential applications in flexible electronics.

Freestanding single-crystalline oxide membranes now attract much attention for their extraordinary mechanical and electrical performance, as compared to conventional thin films on rigid substrates[1–3]. Freestanding single-crystalline oxide membranes are usually prepared by epitaxial thin film deposition with the assistance of a sacrificial layer. They could be fully peeled off after removing the sacrificial layer and transferred to arbitrary substrates for heterogeneous adhesion[4]. It is very promising for flexible electronics because we could not deposit single-crystalline oxides on polymer substrates in the past. A lot of freestanding single-crystalline oxide membranes have been studied, such as SrTiO₃[5], BaTiO₃[2] and La₀.₇Ca₀.₃MnO₃[1], for their excellent dielectric, ferroelectric (FE) or ferromagnetic properties. Among them, freestanding single-crystalline oxide FE membranes could exhibit

superior elasticity and flexibility considerable to polymers. For example, the freestanding BaTiO₃[2] and BiFeO₃[6] membranes with a thickness of about one hundred nanometers could endure a bending strain of > 10% and > 5%, respectively. They can be 180° folded, and the radius of curvature is as small as ~1 μm. Such super-elasticity of BaTiO₃ membranes arises from the continuous rotation of polarization and the dynamic evolution of FE nanodomains[2]. For BiFeO₃ membranes, it mainly originates from reversible rhombohedral-tetragonal phase transition[6]. With the rapid development of wearable electronic devices and smart sensors, flexible electronics require integrating more and more freestanding single-crystalline oxide membranes with multi-functionality.

Antiferroelectric (AFE) oxides are multifunctional materials that have attracted much attention due to their unique field-induced phase

[1]State Key Laboratory for Manufacturing Systems Engineering, Electronic Materials Research Laboratory, Key Laboratory of the Ministry of Education, School of Electronic Science and Engineering, Xi'an Jiaotong University, Xi'an 710049, China. [2]School of Environmental and Material Engineering, Yantai University, Yantai 264005, China. [3]State Key Laboratory for Mechanical Behavior of Materials, Xi'an Jiaotong University, Xi'an 710049, China. [4]These authors contributed equally: Yunting Guo, Bin Peng, Guangming Lu. ✉e-mail: pengbin@xjtu.edu.cn; lisuzhi@xjtu.edu.cn; mingliu@xjtu.edu.cn

transition[7,8]. The field-induced strain could reach 1.1% during the AFE-FE phase transition[9], much larger than the piezoelectric strain in most of the ferroelectric materials. They also exhibit a large electric polarization response at the high electric field and weak remnant polarization at the zero field. Those characteristics make them suitable for a wide range of applications in actuation, energy storage[10–12], memory devices[13], solid-state refrigeration[14] and thermal switches[15], etc. These extraordinary performances are usually achieved in AFE single-crystals or AFE epitaxial thin films. To meet the demand for next-generation flexible electronic devices, it is necessary to manufacture AFE single-crystalline oxides as flexible as metals and polymers. Traditionally, the widely studied flexible AFE oxide thin films are those deposited directly on metal foils[16,17] and mica[11–13,18] substrates. However, they either have poor crystallinity or their flexibility is constrained by the substrate, with a maximum bending strain far below 1%[11,16,19]. The fabrication of freestanding single-crystalline oxide membranes is an attractive method to achieve both perfect crystallinity and superior flexibility at the same time. Recently, we fabricated freestanding single-crystalline PbZrO₃ AFE membranes and studied their energy storage performance in the "organic/inorganic" composite[20]. However, the elasticity of freestanding single-crystalline AFE membranes and the fundamental mechanism have not been explored yet.

In this study, we choose freestanding single-crystalline PbZrO₃ membranes, a classic AFE material, as the prototype to study the effect of internal lattice strain and external strains on its dielectric and ferroelectric properties. We fabricated freestanding single-crystalline PbZrO₃ membranes with a damage-free lifting-off process. We further observed that the freestanding PbZrO₃ membranes could exhibit remarkable flexibility under bending deformation by in situ scanning electron microscopy (SEM). The maximum recoverable strain could reach up to 3.5%. With atomistic simulations, it is revealed that the AFE-FE phase transition is responsible for the shape recovery in our freestanding PbZrO₃ films.

## Results

### Preparation of freestanding PbZrO₃ membranes

Freestanding single-crystalline PbZrO₃ membranes are fabricated by a water-soluble sacrificial layer method, as shown in Fig. 1a. The Sr₃Al₂O₆ was chosen as a sacrificial layer because it is water-soluble and has a good lattice match with PbZrO₃. The PbZrO₃/Sr₃Al₂O₆ heterostructure was epitaxially grown on SrTiO₃ substrates by pulsed laser deposition. Subsequently, a soft supporting layer was either coated onto or affixed to the surface of the as-grown heterostructures, followed by water-etching of Sr₃Al₂O₆ to obtain freestanding PbZrO₃ membranes. Figure 1b shows a millimeter-scale (2.5 mm × 2.5 mm) freestanding PbZrO₃

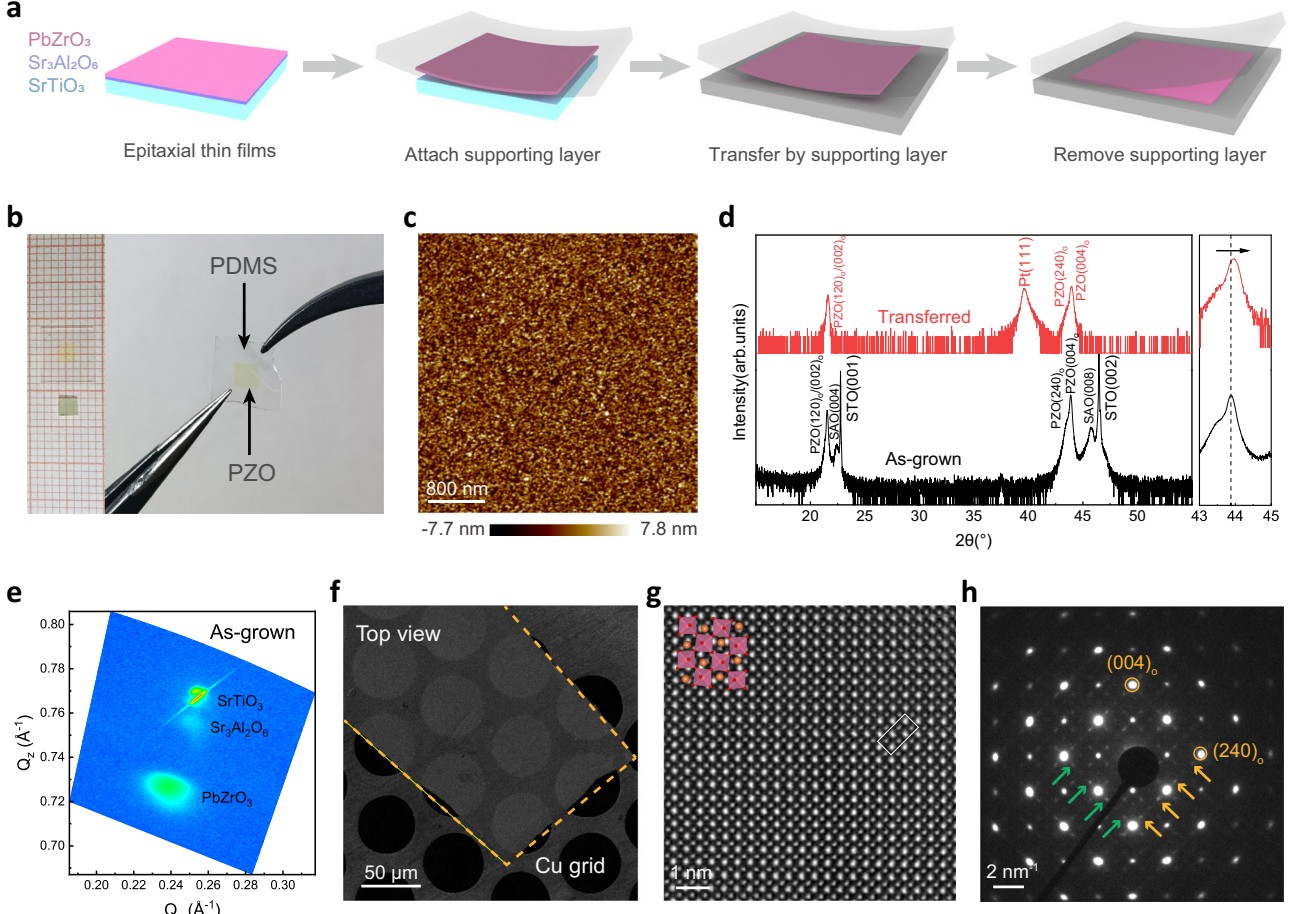

**Fig. 1 | Fabrication of freestanding single-crystalline PbZrO₃ (PZO) membranes.**
**a** Schematics of the whole fabrication process as depositing PbZrO₃/Sr₃Al₂O₆ (SAO) heterostructure on SrTiO₃ (STO) substrates, releasing PbZrO₃ films from the substrates by water etching, and then transferring PbZrO₃ membranes to other substrates. **b** Optical images of a 2.5 mm × 2.5 mm PbZrO₃ membrane transferred to polydimethylsiloxane (PDMS) with good flexibility. **c** Surface morphology of transferred PbZrO₃ on platinized silicon substrate scanned by AFM. **d** X-ray diffraction patterns of as-grown SrTiO₃/Sr₃Al₂O₆/PbZrO₃ heterostructure and freestanding PbZrO₃ membranes on the platinized silicon substrate. a.u., arbitrary units. **e** The RSM studies of as-grown SrTiO₃/Sr₃Al₂O₆/PbZrO₃ heterostructure around (103) diffraction. **f** Top-view TEM image of a freestanding PbZrO₃ membrane supported by Cu grid with lacey carbon film. **g** Atomic-resolution HAADF-STEM image of a freestanding PbZrO₃ membrane from the top view. **h** Selected-area electron diffraction pattern of freestanding PbZrO₃ membrane.

membrane transferred to a polydimethylsiloxane (PDMS) substrate. They are highly flexible and remain intact. Those freestanding membranes and transferred ones are crack-free, as observed by SEM with high magnification (Supplementary Fig. S1). The atomic force microscopy (AFM) image reveals that freestanding PbZrO$_3$ membranes have a very smooth surface with a roughness of 2.26 nm (Fig. 1c).

The single-crystalline structure of freestanding PbZrO$_3$ membranes is confirmed by both X-ray diffraction (XRD) and transmission electron microscopy (TEM). Bulk PbZrO$_3$ has an orthorhombic structure at room temperature with lattice constants of $a_o = 5.884$ Å, $b_o = 11.768$ Å, and $c_o = 8.220$ Å, which can also be represented as a simplified pseudotetragonal perovskite unit cell with lattice constants of $a_t = b_t = 4.161$ Å, and $c_t = 4.110$ Å[21]. Since the cubic Sr$_3$Al$_2$O$_6$ has a lattice constant of $a_c = 15.844$ Å and its $a_c/4$ is smaller than $a_t$ of PbZrO$_3$, the PbZrO$_3$ film grown on Sr$_3$Al$_2$O$_6$ could be subjected to in-plane compressive stress. Figure 1d compares the XRD profiles of as-grown heterostructures and the freestanding PbZrO$_3$ membranes. We can see that the (004)$_o$ peak of the freestanding PbZrO$_3$ sample shifts to the right in comparison to that in the as-grown sample, indicating that the contraction of out-of-plane lattice spacing for PbZrO$_3$ is estimated to be −0.23% (for calculations, see Supplementary Table S1). Figure 1e shows the reciprocal space mapping (RSM) images of as-grown SrTiO$_3$/Sr$_3$Al$_2$O$_6$/PbZrO$_3$ heterostructure around (103). The bright diffraction spots indicate good single-crystallinity and epitaxy relationship of PbZrO$_3$ on SrTiO$_3$ substrates. The RSM of the freestanding PbZrO$_3$ membrane (Supplementary Fig. S2) shows that the in-plane lattice constant of PbZrO$_3$ increases, indicating the in-plane compressive strain in PbZrO$_3$ is released. The epitaxy relationship of SrTiO$_3$/Sr$_3$Al$_2$O$_6$/PbZrO$_3$ heterostructure was also confirmed by the cross-sectional TEM image (Supplementary Fig. S3). From the atomic resolution high-angle annular dark-field scanning TEM (HAADF−STEM) image, the interface between Sr$_3$Al$_2$O$_6$ and PbZrO$_3$ is sharp and clear.

The single-crystalline structure of freestanding PbZrO$_3$ membranes (120-nm-thick) was further confirmed by top-view STEM, as shown in Fig. 1f, which had good electron transparency. Figure 1g shows an atomic-resolution HAADF-STEM image. The selected-area electron diffraction (SAED) patterns are shown in Fig. 1h and more cases are shown in the supplementary Fig. S4. We could observe very weak $\frac{1}{2}\{110\}_c <$ ($c$ denotes pseudo-cubic unit cell) and obvious $\frac{1}{4}\{110\}_c$ super-lattice reflection, which indicate the antiparallel shift of Pb$^{2+}$ cations of AFE membranes[22–24] and the existence of commensurate AFE phases with dipole aligns like ↑↓↑↓ and ↑↑↓↓, respectively. The notable elongation of $\frac{1}{4}\{110\}_c$ along <110> directions as $\frac{1}{n}\{110\}_c$ ($n$ is non-integer) implies the incommensurate AFE phases with dipole arrangement like ↑↑↑↓↓↑↑↑↓↓ and here $n = 3.38 \sim 4.78$. Therefore, commensurate and incommensurate AFE phases coexist in freestanding PbZrO$_3$. The commensurate AFE phases produce zero remnant polarization ($P_r$), while the incommensurate AFE phases results in a non-zero $P_r$.

## Antiferroelectricity of freestanding PbZrO$_3$ membranes under different strains

Freestanding single-crystalline PbZrO$_3$ membranes provide a platform to explore the effect of strain on its electric performance. Firstly, we examined the effect of internal bi-axial lattice strain on dielectric and antiferroelectric behavior. Figure 2a shows the field-dependent dielectric permittivity ($\varepsilon$) and dielectric loss ($\tan\delta$) of as-grown SrTiO$_3$/Sr$_3$Al$_2$O$_6$/SrRuO$_3$/PbZrO$_3$ and freestanding SrRuO$_3$/PbZrO$_3$ thin films. Here, an ultrathin SrRuO$_3$ is selected as the bottom electrode. We observed four peaks in electric field-dependent dielectric permittivity curves. They all correspond to AFE-FE transitions, similar to classic antiferroelectric materials. The PbZrO$_3$ membranes/films are antiferroelectric at zero field. When a positive electric field was applied, the first peak emerged at about 291 kV/cm, corresponding to a typical AFE → FE transition. PbZrO$_3$ transforms to the ferroelectric state

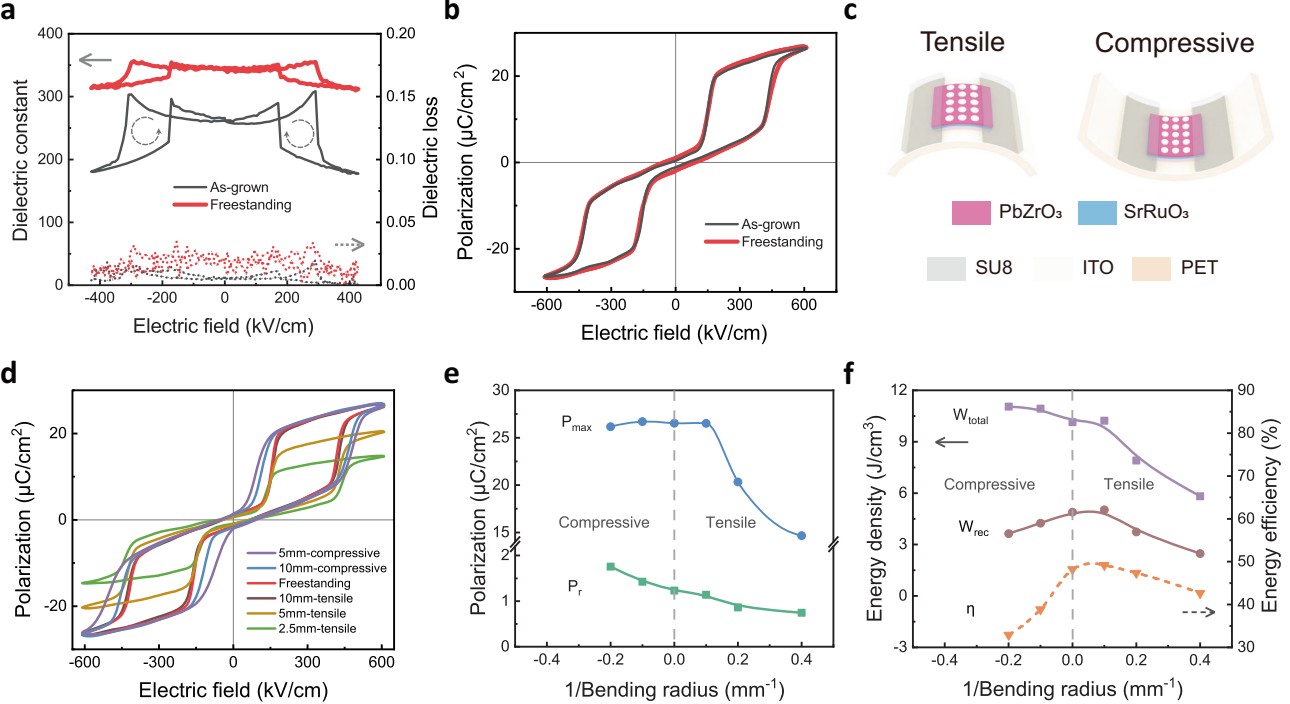

**Fig. 2 | Dielectric and ferroelectric properties of freestanding PbZrO$_3$ membranes.** Electric field dependent dielectric permittivity and loss **a** and *P-E* hysteresis loops **b** of as-grown SrTiO$_3$/Sr$_3$Al$_2$O$_6$/SrRuO$_3$/PbZrO$_3$ heterostructure and free-standing SrRuO$_3$/PbZrO$_3$ membranes on the indium tin oxide (ITO) coated polyethylene terephthalate (PET) substrate (the dotted arrows indicate the scanning sequence). **c** Schematic illustration of bending test with negative photoresist (SU8) as a fixing layer at both sides. **d** The *P-E* hysteresis loops of the freestanding PbZrO$_3$ membranes on the ITO-coated PET substrate during bending. **e** The variation of maximum polarization ($P_{max}$) and remnant polarization ($P_r$) as a function of 1/bending radius. **f** The dependence of total energy density ($W_{total}$), recoverable energy density ($W_{rec}$), and energy storage efficiency ($\eta$) on the 1/bending radius.

at the higher electric field. Another peak starts to appear at about 171 kV/cm during the removal of the applied electric field, corresponding to FE → AFE transition. Similar behavior could be observed when applying/removing a negative electric field. The freestanding membrane has a larger $\varepsilon$ than the as-grown film, implying an enhanced ability to store electrostatic energy. The increase of dielectric constant in freestanding PbZrO$_3$ membrane is possibly due to the removal of the substrate clamping effect. Enhancement of dielectric properties (dielectric constant and polarization) after reduce or fully eliminate such clamping effect is widely observed in ferroelectric films[25]. The as-grown PbZrO$_3$ films are exposed to elastic strain due to lattice mismatch, which constrains the displacement of dipoles pronouncedly under the electric field. Such lattice strain will be released when they become freestanding, as confirmed by XRD and RSM. This is also manifested by the obvious increase of remnant polarization from 0.76 $\mu C/cm^2$ to 1.49 $\mu C/cm^2$, as shown in Fig. 2b. Thus, both the as-grown and freestanding PbZrO$_3$ films exhibit good AFE behavior[26], and the small but non-negligible $P_r$ indicates the existence of ferroelectric-like phase in PbZrO$_3$. Similar behavior has been widely observed in PbZrO$_3$-based antiferroelectric ceramics and films[27,28]. We have examined their $P$-$E$ hysteresis loops at the higher applied electric field ($E > 1000$ kV/cm), as shown in Supplementary Fig. S5. They still exhibit the typical double $P$-$E$ loops for antiferroelectric materials. When $E > 380$ kV/cm, the membranes turn into ferroelectric states and show a nearly linear $P$-$E$ relationship like classic ferroelectric materials, from which we could extract the spontaneous polarization $P_s = 25.5\,\mu C/cm^2$, very close to previous observations[21,29,30].

We also investigated the influence of the external bending strains on the antiferroelectric behavior of freestanding PbZrO$_3$ membranes. They are transferred on the flexible indium tin oxide (ITO) coated polyethylene terephthalate (PET) substrate and attached to different molds with different radii as 10 mm, 5 mm and 2.5 mm (see Supplementary Fig. S6), as shown in the inset in Fig. 2c. The corresponding maximum strains are about 0.83%, 1.66% and 3.3%, respectively (see Supplementary Fig. S7). This strain is much larger than previous

reports applied to metal foils or mica substrates[11,16,19] (see Supplementary Table S2). It should be noted that this is a nominal strain estimated by Eq. S1 and will be influenced by the exact value of mechanical properties of the flexible substrates and membranes. In order to improve the transfer efficiency of the strain, we patterned a fixing layer at both sides of the freestanding membrane after transferring it to PET. Figure 2d (and Supplementary Fig. S8) shows the $P$-$E$ hysteresis loops of the freestanding PbZrO$_3$ membranes on the ITO-coated PET substrate at various bending radii of curvature ($R$). The corresponding maximum polarization ($P_{max}$) and $P_r$ change with the reciprocal of tensile and compressive bending radius are shown in Fig. 2e. When the membrane is under a large tensile strain for the case of $R = 2.5$ mm ($1/R = 0.4$ mm$^{-1}$), the $P_{max}$ reduced significantly because the tensile stress rotates the polarization towards the film plane. When compressive strain is applied to the membrane, the $P_r$ increases, whereas it decreases when tensile strain is applied. The switching electric field ($E_{FE-AFE}$) moves towards the zero field under a compressive strain, indicating the membrane transforms from an antiferroelectric state to a ferroelectric-like state[31–33]. Figure 2f shows the total energy density ($W_{total}$), recoverable energy density ($W_{rec}$), and energy storage efficiency ($\eta$) values at different bending states. Both tensile and compressive strains reduce $W_{rec}$ and $\eta$.

## Elasticity and flexibility of freestanding PbZrO$_3$ membranes

We further examined the shape recoverability of our freestanding single-crystalline PbZrO$_3$ membranes under bending deformation. Figure 3 shows the SEM images of a freestanding PbZrO$_3$ nanoribbon in in-situ bending tests. This nanoribbon was cut by the focused ion beam (FIB). It has a lateral size of 14 μm × 2.5 μm with a thickness of 120 nm (Supplementary Fig. S9a). One end of this nanoribbon was fixed on a nano-manipulator tip (Supplementary Fig. S9b). We used another tip to push the nanoribbon to bend. This allows us to bend the PbZrO$_3$ nanoribbon to a very small radius of curvature, which is down to the micrometer level, three orders of magnitude smaller than that in bending flexible substrate (Fig. 2c). We have bent this nanoribbon

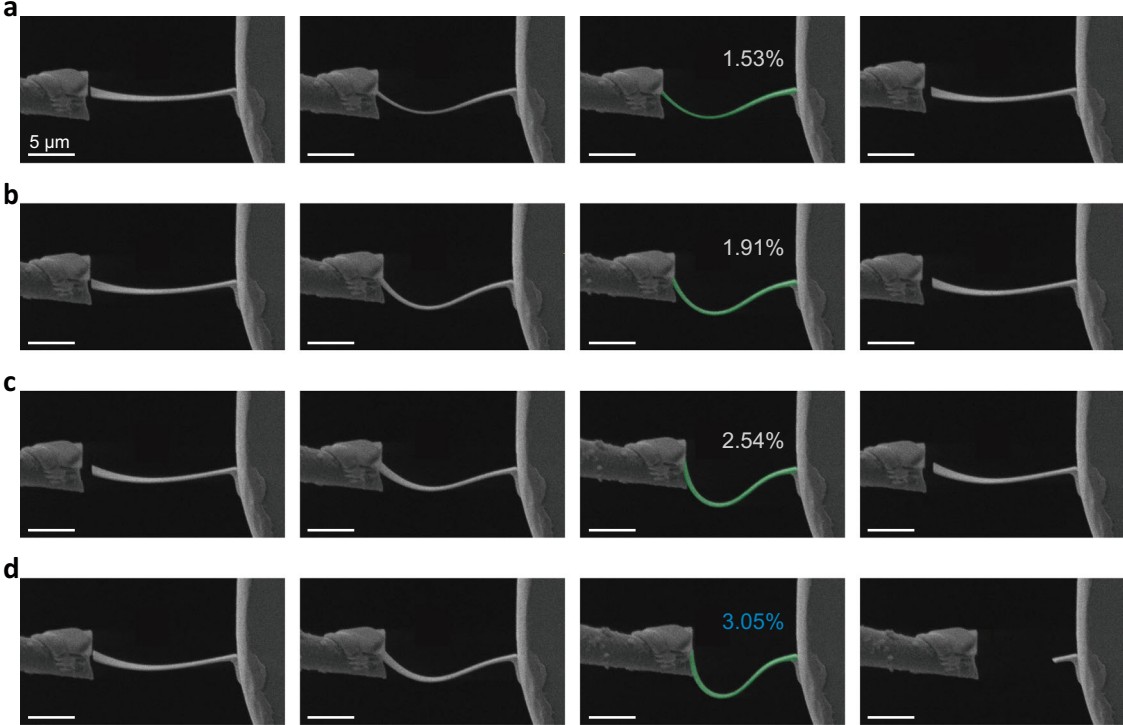

**Fig. 3 | In situ SEM bending test of freestanding PbZrO$_3$ nanoribbon. a–d** The first to fourth columns respectively correspond to the initial, intermediate, maximum and residual bending states during four subsequent bending cycles. Scale bars, 5 μm.

several times with different maximum strains, as shown in Fig. 3 and Supplementary Movie S1. Here, we present the initial, intermediate, maximum and residual bending states of each cycle. During the first cycle (Fig. 3a), the nanoribbon was pushed from left to right, reaching a maximum strain of about 1.53% ($R = 3.93$ μm, calculations see Supplementary Table S3, the strain-radius curve see Supplementary Fig. S10). The nanoribbon could return to its original shape after removing the pushing tip. During the second cycle (Fig. 3b), the maximum bending strain reached 1.91% ($R = 3.14$ μm). Still, after removing the external load, the sample shows good shape recoverability with no residual strain. Even after the third cycle (Fig. 3c) with the maximum bending strain increasing to 2.54% ($R = 2.36$ μm), only a small residual strain was obtained. When the sample is bent to a large strain of 3.05% ($R = 1.97$ μm, see Fig. 3d and Supplementary Fig. S11), it shows a brittle failure. We have confirmed such remarkable flexibility in freestanding $PbZrO_3$ membranes with different film thicknesses (Supplementary Figs. S12 to S14). An even larger maximum bending strain (~3.57%) is achieved in a thicker membrane (~133 nm) (Supplementary Fig. S14).

### Atomistic simulations of freestanding $PbZrO_3$ membranes under different strains

We carried out atomistic simulations to understand the microstructure evolution under different strains in $PbZrO_3$ membranes. Figure 4a shows the typical snapshots of $PbZrO_3$ nanodomains under uniaxial tension and compression, i.e., from the maximum tensile strain of $\varepsilon_{xx} = 5\%$ to the maximum compressive strain of $\varepsilon_{xx} = -5\%$. For the initial configuration, it contains antiparallel displacement of $Pb^{2+}$ ions and uncompensated modulations of polarization (ferroelectric-like regions) at the nanoscale[9]. The dipoles are primarily along four equivalent main directions [101], [−10-1], [−101] and [10-1]. When the in-plane uniaxial compressive strain is applied to the film, the distance between adjacent dipoles gradually decreases, inducing a strong interaction between dipoles. Consequently, dipoles rotate towards the out-of-plane [001] direction (see Supplementary Fig. S15). As marked by the dotted box in Fig. 4a, as the compressive strain increases to −5%, the domain patterns gradually change from an AFE arrangement to a FE arrangement, forming 180° stripe nano-domains with alternating dipoles in [001] and [00-1] directions (Fig. 4c). The thickness of ferroelectric domains could reach ~8 unit cells. When an in-plane uniaxial tensile strain is applied to the film, all dipoles are prone to arrange horizontally along [100] and [−100] directions (see Supplementary Fig. S16). As tensile strain becomes 5%, the vertical anti-parallel dipoles switch to be parallel. The stripe domain with a higher modulation period along the in-plane polarization is formed. Correspondingly, a large ferroelectric domain is formed, indicating a typical AFE-FE transition at local sites under tension (Fig. 4c). In comparison to compression, tensile deformation could afford more space for small domains to evolve and merge. Figure 4e shows the macroscopic polarization density as a function of applied strain. We found that the polarization along the tension direction ($P_x$) increases with increasing external strain. The polarization along the $z$ direction ($P_z$) is almost unchanged, keeping in accordance with experimental observations (see Fig. 2e).

To understand the mechanism governing the high flexibility in our freestanding $PbZrO_3$ membranes, we further calculated the evolution of dipole configurations during bending tests, as shown in Fig. 4b and Supplementary Fig. S17. The maximum bending angle applied to the $PbZrO_3$ membrane is 10°, corresponding to a maximum tensile and compressive strain of 5% at the top and bottom surfaces, respectively. Supplementary Fig. S18 shows the maximum bending strain $\varepsilon_{max}$ as a function of the bending angle $\theta$ in this $PbZrO_3$ membrane. As the bending strain increases, the dipoles at the top of membrane gradually rotate to the tangential direction, similar to the pattern formed at the 5% uniaxial tensile strain (Fig. 4a), while those at the bottom of membrane rotate to the radial direction, close to the state at a −5% uniaxial

compressive strain (Fig. 4a). With the configurations of dipole, we could clearly see that the sample undergoes an obvious AFE-FE transition under the external stress. Between compressive and tensile regions, a transitional region where polarization rotates continuously is formed. Moreover, these regions form a vortex-like domain structure, as marked in the dotted box of Fig. 4d. The formation of this vortex-like structure arises from the gradient strain under bending deformation. As the sample experiences a transition from compressive to tensile strain near the neutral layer, the dipoles are then driven to rotate towards the directions normal to the neutral layer. As a result, a fully closed vortex-like domain structure is generated near the neutral layer. We further calculated the remnant polarization density $P$ and its two components along tangential ($P_{tangential}$) and radial ($P_{radial}$) directions. We observed that the remnant polarization $P$ increases with increasing bending strain (Fig. 4f). Therefore, the highly flexible behavior of $PbZrO_3$ membranes mainly originates from the AFE-FE phase transition with the aid of dipole rotation.

## Discussion

In this study, the remarkable flexibility and elasticity of freestanding $PbZrO_3$ membranes mainly originates from bending-induced AFE-FE transition. Several studies have shown such transition driven by the mechanical load. Chaudhuri et al.[34] and Gao et al.[24] found that the interfacial compressive strain could stabilize the FE phase in $PbZrO_3$ or $PbZrO_3$-based epitaxial thin films, as observed by TEM directly. Similar to our observations (Fig. 2d, e), remanent polarization $P_r$ has a notable increase in the double $P$-$E$ hysteresis loops[34]. This behavior becomes more obvious in ultrathin $PbZrO_3$ films because a much larger compressive strain is generated[34]. Although the microstructure of $PbZrO_3$-based AFE materials is very complex[24,26,34–36], except commensurate modulated structures, in most cases they also have incommensurate modulated structures with non-integral modulation period ($n$), revealed by $\frac{1}{n}\{110\}_c$ diffraction patterns[24]. In this work, the dipole alignment is likely to be a mixture of ↑↑↓↑↑↓ ($n = 3$), ↑↑↓↓↑↑↓↓ ($n = 4$) and ↑↑↑↓↓↑↑↑↓↓ ($n = 5$) with $n$ ranges between 3.38 ~ 4.78 (Fig. 1h). The formation of such a commensurate/incommensurate coexisted modulation arises from the competition between long-range FE orders (parallel alignment of dipoles like ↑↑↑↑) and short-range AFE orders (usually antiparallel alignment of dipoles like ↑↑↓↓ with $n = 4$ for a classic AFE $PbZrO_3$)[24]. Dipole alignment in both commensurate and incommensurate modulated $PbZrO_3$ could be considered as stripe-like 180° FE domains[35]. This incommensurate modulated microstructure takes a dominant role in completing the AFE-FE transition. We found the bending deformation could induce an increase of thickness in such 180° FE domains like that in chemically doped $PbZrO_3$[26], as revealed by our simulations that $n$ increases to 7 ~ 8 (Fig. 4c).

Usually, the maximum bending strain for bulk ceramics is only 0.2% ~ 0.4%[37,38]. Here, it is far beyond 1% for freestanding $PbZrO_3$, which is one order of magnitude larger than the bulk ceramics. Therefore, the present observation in a classic AFE $PbZrO_3$ membranes further extended the family's ability to exhibit giant elasticity and flexibility of single-crystalline perovskite oxides, in combination with previously reported systems of $BaTiO_3$[2,39] (a classic FE material), $BiFeO_3$[3,6,40] (a typical multiferroic material) and PMN-PT[41,42] (a typical relaxor FE material). Although the elastic strain in $PbZrO_3$ is relatively smaller than those materials (>5%), it is still considerable to and even larger than some metallic nano-materials (2% ~ 4%)[43]. Fundamentally, dipole switching, especially the continuous dipole rotation upon a strain gradient, plays a critical role in the shape recovery of bent ferroic oxide. This continuous rotation could largely eliminate the mismatch stress caused by the abrupt change of microstructures at high strain levels, like the AFE-FE transition here, avoiding possible mechanical failure. For $BaTiO_3$ membranes, a transition zone with continuous dipole rotation forms near the neutral plane under bending[2], reducing

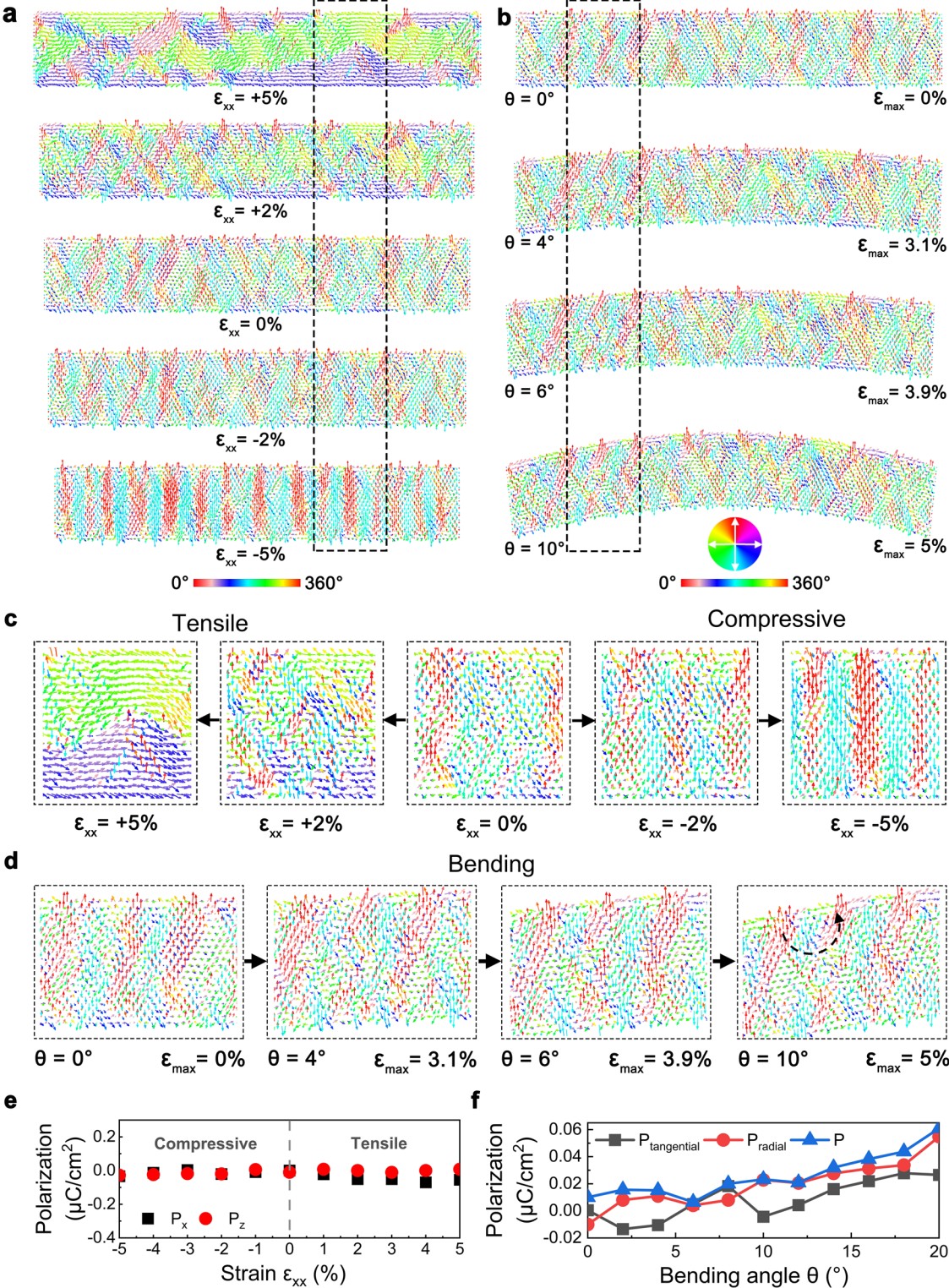

**Fig. 4 | Domain evolution of freestanding single-crystalline PbZrO₃ membrane under the uniaxial strains obtained by atomistic simulations. a** Typical snapshots of dipole configurations under the uniaxial tensile and compressive strains $\varepsilon_{xx}$. **b** Typical snapshots of dipole configurations during the bending process. The $\varepsilon_{max}$ and $\theta$ refer to the maximum bending strain and the bending angle in PbZrO₃ membrane, respectively. **c** The typical configurations at the local site under tension and compression, as the dashed frame marked in (a). **d** The typical configurations at the local site upon bending, as the dashed frame marked in (b). **e** The in-plane polarization $P_x$ and out-of-plane polarization $P_z$ as a function of the applied strain. **f** The polarization density $P$ as a function of the bending angle. The $P_{tangential}$ and $P_{radial}$ refer to the components along tangential and radial directions, respectively.

the lattice stress in the coexisting $c$ and $a$ nanodomains. The super-elasticity of freestanding BiFeO$_3$ membranes originates from rhombohedral-tetragonal phase transition, together with a rather thicker transitional zone where polarization rotates continuously between the thickness direction and diagonal direction of a pseudo-cubic unit cell[6].

In addition, continuous dipole rotation in AFE PbZrO$_3$ could induce other unique dipole configurations. It has recently been reported that both the magnitude and the relative angle between neighboring 180° FE domains may be different and could be accommodated by the ferrielectric (FiE) phase[26,36]. The FiE phase could behave in either the magnitude modulation mode or the angle modulation mode. In our simulations, we found both modes exist at local sites. Bending deformation may promote the formation of such FiE phase with uncompensated polarization as observed experimentally (Fig. 2e). Here for the freestanding PbZrO$_3$, we observed diffraction patterns in both [011]$_c$ and [01-1]$_c$ directions (Fig. 1h), indicating 90° alignment of 180° FE domains and may favor a closed domain structure near the boundaries. Our atomistic simulations further revealed the formation of a vortex-like domain structure near the neutral layer with the aid of dipole rotation under the strain gradient (Fig. 4d). The vortex-like domains could take an important role in assisting the bending-induced high flexibility. Besides, dipole rotation also induces small non-180° FE domains by ferroelastic domain switching and we also observed the ferroelastic domain switching coupled with the electric dipole in our simulations (see the strain map in the bent sample in Supplementary Fig. S19).

PbZrO$_3$ film as well as its chemically doped ones have already exhibited ultrahigh energy-storage density for high-power electro-static capacitors[44] and record-high field-induced strain ($>1\%$)[9] for actuators, but those films are clamped on rigid substrates. Our highly flexible freestanding single-crystalline PbZrO$_3$ membranes now provide a platform for developing advanced flexible devices as well as heterogeneous integration with other substrates. For flexible electronics, it should be noted that the electrical properties of undoped PbZrO$_3$ seem vulnerable to bending strain (Fig. 2d–f). One solution is to embed the membranes in the organic matrix to form a sandwiched composite[20] and keep the freestanding membrane in the center and near the neutral plane. In this way, we could obtain a very stable polarization response during cyclic bending[45]. On the other hand, bending strain-sensitive AFE-FE transition could be further exploited to manipulate electrical and mechanical properties and develop multifunctional devices[46,47]. For example, we could develop a self-rolling-up micro-tube or micro-spring by freestanding single-crystalline oxide bilayer membranes, driven by their internal lattice strain[46,47].

In summary, we have fabricated freestanding single-crystalline PbZrO$_3$ membranes with varied film thickness. The incommensurate modulated AFE structure is formed with different non-integral modulation periods. The PbZrO$_3$ membranes exhibit double hysteresis loops with a small remanent polarization ~0.95 $\mu$C/cm$^2$. Their nanoribbons could endure cyclic bending and possess good shape recoverability for a bending strain of ~2.5%. The tolerated maximum bending strain could reach 3.57%. Atomistic simulations revealed that this remarkable flexibility originates mainly from the AFE-FE transition accomplished by the continuous dipole rotation upon bending. The freestanding PbZrO$_3$ membranes provide a useful platform to develop advanced flexible AFE devices.

## Methods
### Epitaxial thin film deposition
The fabrication of Sr$_3$Al$_2$O$_6$ and PbZrO$_3$ thin films on SrTiO$_3$(001) substrates was achieved through a series of controlled deposition steps using pulsed laser deposition, employing a KrF excimer laser with a wavelength of 248 nm. Initially, the Sr$_3$Al$_2$O$_6$ layer was deposited at a growth temperature of 760 °C, under an oxygen pressure of 15 Pa,

with an energy density of approximately 0.8 J/cm$^2$ and a pulse repetition frequency of 3 Hz. Following this, the PbZrO$_3$ layer was deposited at a temperature of 575 °C, under an oxygen pressure of 20 Pa, utilizing an energy density of about 0.5 J/cm$^2$ and a pulse repetition frequency of 5 Hz. Subsequently, the SrTiO$_3$/Sr$_3$Al$_2$O$_6$/PbZrO$_3$ hetero-structures were subjected to in situ annealing within the deposition chamber at 575 °C and an oxygen pressure of approximately 90 kPa for 20 min. For electrical measurement, electrodes composed of either Au or Pt, each with a diameter of 50 $\mu$m and a thickness of 50 nm, were uniformly deposited onto the film surface. The creation of point electrodes involved a photolithographic process, followed by their deposition using a magnetron sputtering system. This was carried out under a stringent vacuum environment, with a base pressure of less than $1 \times 10^{-7}$ Torr, a working argon pressure of 3 mTorr, and a DC power setting of 50 W. The thickness of these electrodes was meticulously calibrated using a quartz crystal microbalance, which is an integral component of the sputtering system, ensuring precise control over the electrode thickness.

### Release and transfer PbZrO$_3$ membranes
A soft supporting layer, which could be polydimethylsiloxane (PDMS), Polyimide (PI), or photoresist (AR-P 3510 T, Allresist GmbH), was applied or affixed to the surface of PbZrO$_3$ epitaxial thin films. The films, with the supporting layer, were then submerged in deionized water at ambient temperature. This process continued until the Sr$_3$Al$_2$O$_6$ layer was fully dissolved, enabling the separation of the PbZrO$_3$ membrane from the substrates. The use of PDMS or PI as the supporting layer provided the added advantage of serving as flexible substrates. To transfer the PbZrO$_3$ membranes onto different substrates, such as silicon or PET, the "photoresist/PbZrO$_3$" stack was initially transferred onto the target substrate. Following this, the photoresist layer was thoroughly removed using acetone, ensuring a clean and secure adhesion of the PbZrO$_3$ membrane to the new substrate.

### Microstructure characterization of membranes
The surface topography of the samples was examined using an atomic force microscope (AFM, Bruker, Dimension Icon). Additionally, high-resolution x-ray diffraction (XRD) measurements, specifically θ-2θ scans, were performed using a PANalytical Empyrean diffractometer equipped with a Cu Kα1 radiation source to analyze and confirm the crystallographic properties of the materials. Furthermore, atomic-resolution high-angle annular dark-field (HAADF) images were captured using a JEOL ARM200F microscope, which was outfitted with a corrective spherical aberration (CS) STEM operating at an accelerating voltage of 200 kV. The in-situ scanning electron microscopy (SEM) bending tests on PbZrO$_3$ nanoribbons were conducted using a FIB (Helios NanoLab DualBeam) system, equipped with Kleindiek mechanical manipulators. The fabrication of PbZrO$_3$ nanoribbons was achieved from large-area freestanding membranes by employing a FIB system with Ga+ sources and low ion current.

### Electrical measurement
The dielectric characteristics were precisely assessed using a high-precision LCR meter (E4980A, Keysight). To further investigate the antiferroelectric properties, a specialized ferroelectric analyzer (aix-ACCT TF2000) was utilized. After transferring the freestanding PbZrO$_3$ membrane on the flexible ITO coated PET substrate, the negative photoresist (SU8) was spun-coated on both sides of it as a fixing layer (3000 rpm, 50 s). And then attached it to molds with different radii using tape or double-sided tape to test its antiferroelectric properties.

### Atomistic simulations
The interatomic interactions of PbZrO$_3$ were described by a core-shell model[48], which consists of positively charged shell and

negatively charged core for each atom. The whole potential energy is the sum of the following three parts as the short-range interaction between shells in a Buckingham potential, the short-range anharmonic interaction between core and shell, and the long-range Coulombic interaction[49–51]. Some basic properties, such as ferroelectric aging[52], electrocaloric effect at ferroelectric-antiferroelectric phase boundary[53], high-temperature ferroelectric domain structure[54] and piezoelectric effect[55] could also be reproduced with this potential. The paraelectric $PbZrO_3$ membrane was initially created at 600 K along $x$-[100], $y$-[010] and $z$-[001]. Here we take a quasi-2D model with one lattice unit in $z$ direction to reduce the high computing cost for calculating the long-range Coulombic interactions. This membrane has 80 nm in $x$ direction and 8 nm in $y$ direction. After relaxation, it was cooled down to 300 K. A paraelectric phase to antiferroelectric phase transition occurs at around 420 K with the generation of typical antiferroelectric domain structures. Truncated boundary conditions were adopted in $x$ and $y$ directions while periodic boundary condition was applied in thickness $z$ direction. Several atomic layers at two ends of the membrane in the $x$ direction were fixed rigidly as the loading grid to apply external strain. Tensile, compression and bending deformations were achieved in a displacement-controlled method. For each time with a prescribed strain, the sample was relaxed at 300 K for 100 ps using a Nosé-Hoover thermostat[56]. The trajectory of atom in the last 50 ps was averaged for calculating the atomic strain, diploe configuration etc. We obtained the whole polarization density as $P = qs/V$, where $q = 2.95 \times 10^{-19}$ is the charge of Zr, $s$ is the net displacement and $V$ is the volume of the entire system. All the calcuations were carried out with the DL_POLY code[57]. The atomic configurations were displayed using OVITIO[58].

### Reporting summary

Further information on research design is available in the Nature Portfolio Reporting Summary linked to this article.

## Data availability

The data supporting the findings of this study are available within the main text, the Supplementary Information file, the Source Data files, or from the corresponding authors upon request. Source data has been deposited in Figshare (https://doi.org/10.6084/m9.figshare.25397959) and also provided with this paper.

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

## Acknowledgements

This work was supported by the Natural Science Foundation of China (grants 62131017 [M.L.], U22A2019 [M.L.], 62371387 [B.P.], 12304130 [G.L.], 51931004 [X.D. and J.S.]), the National Key R&D Program of China (2019YFA0307900 [S.L.]), Key R&D Project of Shaanxi Province-University Joint Project (2023GXLH-020 [M.L.]) and the Fundamental Research Funds for the Central Universities [M.L.]. We appreciate for the support in the TEM and SEM tests from the Analysis and Testing Center of the Xi' an Jiaotong University.

## Author contributions

B.P. and M.L. initialized the idea and supervised the project. Y.G. designed the experiments, fabricated samples and conducted electrical measurements. G.L. carried out the atomistic simulations under the guidance of S.L., X.D. and J.S. G.D. and G.Y. conducted TEM and SEM bending tests. R.Q., H.L., B.Z., Y.Y. and B.C. helped prepare samples and characterize their micro-structure. Y.G., B.P. and G.L. wrote the draft with help from M.L., S.L. and Y.Z. in the revision. All authors contributed to the discussion of the results and the revision of the manuscript.

## Competing interests

The authors declare no competing interests.
