## [Peer Review File · Nature Communications]

Remarkable Flexibility in Freestanding Single-crystalline Antiferroelectric PbZrO₃ MembranesEditorial Note: Parts of this Peer Review File have been redacted as indicated to remove third-party material where no permission to publish could be obtained.

REVIEWER COMMENTS

Reviewer #1 (Remarks to the Author):

Guo et al reported an investigation of superior elasticity in freestanding single-crystalline antiferroelectric PbZrO₃ membranes. The polarization responses under strain, compression or bending were carefully studied, with the assistance of simulation. The work provides clear insights into the deformation response and mechanism of AFE oxides. Nevertheless, there are some issues need to be addressed before the publication.

(1) The authors claim the AFE membrane as single-crystalline, which was prepared through pulsed laser deposition on Sr₃Al₂O₆ substrate. However, there is no any evidence that the free-standing membrane is single-crystalline instead of multi-domain structure. Powder XRD gives the average diffraction pattern, while the ED presents the diffraction pattern in rather local area. They can't prove the single-crystalline nature. The authors need to collect the 2D XRD diffraction patterns at various positions of AFE oxide membranes and compare the orientations of diffraction patterns.

(2) In Figure 2A, the scanning sequence of dielectric profiles are suggested mark by arrows.

(3) In Figure 2B, the PbZrO₃ oxide exhibits a typical AFE polarization curve under 600 kV/cm. What's the polarization behavior would be at higher applied electric field, e.g. 1000 kV/cm? Would FE polarization loop be detected? Please check this.

(4) The resolution of Figure 4 is not high enough, which makes it hard to recognize the dipole vector direction. At the compression strain of -5%, the dipole configuration still looks like antiferroelectric since the dipoles are dispersed antiparallel, although the size of each dipole domain grows bigger. It may not be precise to describe the configuration change as AFE-FE transition. On the other hand, at 5%-10% tensile strain, the single domains of parallel dipoles are even larger and present more ferroelectric nature. How do the authors call this dipole configuration change? They are not discussed in the manuscript.

(5) As the authors claim that this super-elasticity originates mainly from the AFE-FE transition, could the authors measure the P-E polarization loop of AFE membrane at high strain? Ferroelectric like profiles should be able to be observed if it's real AFE-FE transition.

(6) Usually the mechanical properties of inorganic membranes are strongly related to the thickness, e.g. strain or elasticity. With this simple preparation method of PbZrO₃ membrane, the authors are suggested to prepare the AFE membranes with various thicknesses and compare their elasticity dependence on membrane thickness.

Reviewer #2 (Remarks to the Author):

In this paper, authors have successfully fabricated a freestanding single-crystalline PbZrO₃ membrane, which is an antiferroelectric oxide. The freestanding membrane shows ultrahigh flexibility. Moreover, the elasticity and antiferroelectricity of freestanding PbZrO₃ membrane are investigated. The experimental results indicate that the freestanding PbZrO₃ membrane can accommodate a large bending strain of up to 3% without any crack or damage. Meanwhile, excellent dielectric and ferroelectric properties of freestanding PbZrO₃ membranes have been observed. This work is interesting, and I would recommend the article for publication after the following points are revised or explained:

1. For Fig. 1g, the author described that “The corresponding selected-area electron diffraction (SAED) pattern shows typical 1/4 superlattice reflections...”. However, though the diffraction spots have been marked by arrows, it is hard to observe the “1/4 superlattice reflections” in the current figure. Maybe, the brightness and contrast could be readjusted to make the SAED pattern more clearly.
2. In Fig. 2a, the curves of field-dependent dielectric permittivity of as-grown and freestanding thin films are significantly different. What is the reason for this difference? The release of strain from the substrate or other reasons? The authors are suggested to take more explanation for this phenomenon. Meanwhile, the author could add some description for four dielectric peaks to make the AFE-FE or FE-AFE transition process more clearly.
3. As we all know, the inorganic ferroelectric thin films could not accommodate too large strain, typically below 2%. In this work, the freestanding thin film has a super elasticity, however the critical strain is about 3%. In the simulation section, the calculated strain is up to 9%, which is much larger than the critical strain of freestanding thin film. I think, it is necessary to explain the relationship between experimental and simulated results and the reason why such a large strain is adopted in the simulations. Moreover, in Fig. 4c, the vortex-like domain structure is observed. How to understand this result? Is it useful for explaining the experimental results?

Reviewer #3 (Remarks to the Author):

The topic of freestanding antiferroelectric PZO is fascinating. The author used a few methods to explore the structure, functionality, and deformability of the PZO membranes. However, the results presented by the authors are insufficient to substantiate the conclusions they have reached. The 3% elastic strain the authors report is not remarkably distinct from that in other perovskite oxides. While the authors suggest that the transition from antiferroelectric to ferroelectric states contributes to elastic deformation, they fail to offer concrete experimental evidence. Additionally, the authors need to address the following questions:

1. On page 3, the authors describe, “Fig. 1b shows a millimetre-scale (2.5 mm × 2.5 mm) crack-free freestanding PbZrO₃ membrane transferred to a polydimethylsiloxane (PDMS) substrate, which still has high flexibility”. To substantiate their claim of no cracking, an SEM image or a high-magnification optical microscopy image is necessary.
2. Regarding perovskite oxides, there are a few studies documenting their exceptional deformability, even in single crystals. Similar behaviours have been reported in membranes, as referenced by the authors. Given that this paper focuses on outstanding deformation ability, it should include more in-depth discussions about the mechanisms of deformation in perovskite oxides.
3. On page 3, the authors state, “The atomic force microscopy (AFM) image reveals that freestanding PbZrO₃ membranes have a very smooth surface with a surface roughness of 2.26 nm (Fig. 1c).” However, based on the scale bar, it is evident that the surface roughness exceeds 2.26 nm.
4. Figure 1(D) on Page 4: The indexing of the XRD peaks appears to be incorrect, particularly for PZO, where the out-of-plane lattice parameter is double that of the cubic perovskite structure. Moreover, there is a shoulder peak for the PZO, which is likely attributable to the orthorhombic (240).

5. Figures 1(F, G) on page 4: The antiparallel displacement of Pb atoms is not visible in the HAADF-STEM image (panel F). Additionally, the $1/4$ superlattice reflections in the FFT image (panel G) are much weaker than those reported in other studies of PZO.
6. The structure of both the as-grown PZO thin film and the free-standing PZO membrane requires more thorough examination. The PZO structure is expected to be complex. Even in the PZO single crystal, a ferrielectric phase may appear.
7. The hysteresis loops depicted in Figure 2(B) appear significantly more complex than typical antiferroelectric loops. The authors should provide an explanation for this occurrence.
8. Figure 2(D): The colors of the different loops appear quite similar. It is suggested to use colors that are distinctly different for better clarity.
9. More details should be provided regarding the bending deformation (Fig. 2(C)), such as the geometry of the mold, how to put the sample on it, how to bend it, etc.
10. Ferroelastic domain switching, which was reported in ferroelectric materials to contribute to elastic deformation, should be considered.
11. In the simulation, it appears the authors did not take into account the impact of ferroelastic switching and intermediate ferrielectric phases. Without experimental evidence, it's uncertain whether these results accurately represent actual conditions.

Response to reviewers

Reviewer #1:

Guo et al reported an investigation of superior elasticity in freestanding single-crystalline antiferroelectric PbZrO₃ membranes. The polarization responses under strain, compression or bending were carefully studied, with the assistance of simulation. The work provides clear insights into the deformation response and mechanism of AFE oxides. Nevertheless, there are some issues need to be addressed before the publication.

Thanks for your careful reading and positive evaluation on the importance of this work. The comments and suggestions would help us to improve the quality of our work.

Comment 1: *The authors claim the AFE membrane as single-crystalline, which was prepared through pulsed laser deposition on Sr₃Al₂O₆ substrate. However, there is no any evidence that the free-standing membrane is single-crystalline instead of multi-domain structure. Powder XRD gives the average diffraction pattern, while the ED presents the diffraction pattern in rather local area. They can't prove the single-crystalline nature. The authors need to collect the 2D XRD diffraction patterns at various positions of AFE oxide membranes and compare the orientations of diffraction patterns.*

Reply: Thanks for your advice. We have conducted reciprocal space mapping (RSM) of the as-grown SrTiO₃/Sr₃Al₂O₆/PbZrO₃ heterostructure and freestanding PbZrO₃ membranes transferred on silicon substrates, as shown in **Fig. R1**. Those RSM results are collected around (103) diffraction peaks of SrTiO₃ substrates. The bright diffraction spots in those RSM indicate very good single-crystallinity and epitaxy relationship of PbZrO₃ on SrTiO₃ substrates. Their in-plane lattice constant of PbZrO₃ increased from 4.159 Å to 4.168 Å and the out-of-plane lattice constant decreased from 4.126 Å to 4.119 Å. We have updated those RSM results and more discussion in the revised manuscript. We have updated these figures and relevant discussion in the revised manuscript.

Fig. R1 (also updated Fig. 1e and newly added Fig. S2) a-b Reciprocal space mapping (RSM) of PbZrO₃ film around (103) peak before (a) and after (b) releasing from the substrates.

Comment 2. *In Figure 2A, the scanning sequence of dielectric profiles are suggested mark by arrows.*

Reply: Thanks for the suggestion. We have updated the dielectric profiles (ϵ - E curves) to illustrate the scanning sequence more clearly, as shown in Fig. R2 (also Fig. 2a in the revised manuscript). When the forward electric field is smaller than the switching electric field (E_{AFE-FE}), the film stays in an anti-ferroelectric state. Both ϵ and $\tan\delta$ increase with increasing electric field. After crossing the critical electric field, the film transforms to a ferroelectric state since the reversal of the adjacent sub-lattice. ϵ and $\tan\delta$ then decrease suddenly. When the reverse electric field decreases to less than the switching electric field (E_{FE-AFE}), the film returns back to the original anti-ferroelectric state.

Fig. R2 (also updated Fig. 2a) Electric field dependent dielectric permittivity and loss of as-grown SrTiO₃/Sr₃Al₂O₆/SrRuO₃/PbZrO₃ heterostructure and freestanding SrRuO₃/PbZrO₃ membranes on the indium tin oxide (ITO) coated PET substrate (the dotted arrows indicate the scanning sequence).

Comment 3. *In Figure 2B, the PbZrO₃ oxide exhibits a typical AFE polarization curve under 600 kV/cm. What's the polarization behavior would be at higher applied electric field, e.g. 1000 kV/cm? Would FE polarization loop be detected? Please check this.*

Reply: We have examined the P - E hysteresis loops at the higher applied electric field ($E > 1000$ kV/cm) of as-grown SrTiO₃/Sr₃Al₂O₆/SrRuO₃/PbZrO₃ heterostructure and freestanding SrRuO₃/PbZrO₃ membranes, as shown in Fig. R3 (also newly added Fig. S5 in supporting information). The PbZrO₃ membranes still exhibit typical double P - E loops for antiferroelectric materials polarization curves even under 1000 kV/cm. The membranes turn into ferroelectric states at the high field and show a nearly linear P - E relationship like classic ferroelectric materials, from which we could extract the spontaneous polarization $P_s = 25.5$ $\mu\text{C}/\text{cm}^2$, very close to previous observations. We have added these new results in the revised Supplementary Information. We have also provided the relevant discussion in the revised manuscript.

Fig. R3 (also newly added Fig. S5) **a** The P - E hysteresis loops at the higher applied electric field of as-grown SrTiO₃/Sr₃Al₂O₆/SrRuO₃/PbZrO₃ heterostructure and freestanding SrRuO₃/PbZrO₃ membranes on the indium tin oxide (ITO) coated PET substrate. **b** The P - E hysteresis loops of freestanding SrRuO₃/PbZrO₃ membranes with the increasing electric field.

Comment 4. *The resolution of Figure 4 is not high enough, which makes it hard to recognize the dipole vector direction. At the compression strain of -5%, the dipole configuration still looks like antiferroelectric since the dipoles are dispersed antiparallel, although the size of each dipole domain grows bigger. It may not be precise to describe the configuration change as AFE-FE transition. On the other hand, at 5%-10% tensile strain, the single domains of parallel dipoles are even larger and present more ferroelectric nature. How do the authors call this dipole configuration change? They are not discussed in the manuscript.*

Reply: We thank the referee for the good comment. We have enhanced the resolution of Fig. 4 for a better view. We also examined the dipole configurations at local sites under mechanical deformation, as indicated by the dashed frame in Fig. R4a. We found that the AFE-FE transition occurs at both tensile and compressive strains. For the uniaxial compression, as the external strain increases, some local antiparallel dipoles switch towards the uniform direction, leading to the formation of typical 180° ferroelectric nano-domains, as shown in Fig. R4b (also newly added Fig. 4c). The thickness of ferroelectric domains could reach ~8 unit cells in x direction at a compressive strain of $\epsilon_{xx} = -5\%$. For tensile deformation, the ferroelectric domains evolve more robustly along the tensile direction (x direction) with the generation of larger ferroelectric patches. When the tensile strain increases to 5%, the vertical anti-parallel dipoles switch to be parallel, as the green dipoles shown in Fig. R4c (also newly added Fig. 4c). Correspondingly, a large ferroelectric domain is formed, indicating a typical AFE-FE transition under tension.

With the analysis above, we could confirm that both the compressive and tensile strains could induce the AFE-FE transitions in the present PbZrO₃ membrane. The ferroelectric nano-domains are formed at local sites to accommodate the external plastic strain. We have revised our manuscript to make this point clear.

Fig. R4 Domain evolution of freestanding single-crystalline PbZrO_3 membrane under the uniaxial strains. a Atomic configuration of the sample under the uniaxial tensile and compressive strains. The colors are coded according to the dipole direction. **b-c (also newly added Fig. 4c)** The typical configuration at the local site under (b) compression and (c) tension, as marked in (a).

Comment 5. As the authors claim that this super-elasticity originates mainly from the AFE-FE transition, could the authors measure the P-E polarization loop of AFE membrane at high strain? Ferroelectric like profiles should be able to be observed if it's real AFE-FE transition.

Reply: As suggested by the reviewer, we have examined the *P-E* hysteresis loops of the freestanding PbZrO₃ membranes with larger bending strain. We transferred the freestanding PbZrO₃ membranes on the PET substrate, which is thicker than the PI substrates we used previously. Now we could apply a much larger bending strain since both the thickness and Young's modulus of PET is larger than PI substrates, as estimated in updated Supplementary Fig. S7. The PET substrates are then attached to molds with different radii as 10 mm, 5 mm and 2.5 mm, corresponding maximum strains are about 0.83%, 1.66% and 3.3%, respectively (see updated Fig. S7). It should be noted that the nominal strain estimated in calculations could be influenced by the exact value of mechanical properties of the flexible substrates and membranes. We could observe the obvious change of remanent polarization during bending. It increases from 0.95 $\mu\text{C}/\text{cm}^2$ to 1.35 $\mu\text{C}/\text{cm}^2$ under compressive strain and decreases to 0.34 $\mu\text{C}/\text{cm}^2$ under tensile strain. Our observations are consistent with previous reports. For example, Chaudhuri *et al.* found that the interfacial compressive strain/stress will stabilize a ferroelectric phase in PbZrO₃ epitaxial thin films as directly observed by TEM (Fig. R6a). Correspondingly, double hysteresis loops turn to ferroelectric-like loops (Fig. R6b and c) with an increase of P_r (Fig. R6d). This is more evident in ultrathin PbZrO₃ films which experience larger compressive strain/stress, while such lattice strain would be released dramatically in thicker films.

Fig. R5 (also updated Fig. 2d-e) a The *P-E* hysteresis loops of the freestanding PbZrO₃ membranes on the ITO-coated PET substrate during bending. b The variation of P_{max} and P_r as a function of $1/\text{bending radius}$.

[Redacted]

Fig. R6 **a** Cross-sectional HRTEM image of SrTiO₃/SrRuO₃/PbZrO₃ heterostructure with $d_{\text{PZO}}=22$ nm. The inset shows selected electron diffraction patterns, indicating the orthorhombic phase (antiferroelectric phase) and the rhombohedral phase (ferroelectric phase). **b** Dynamic *P-E* hysteresis loops for PbZrO₃ epitaxial thin films with different thicknesses. **c** Typical static *P-E* hysteresis loops at room temperature. **d** Thickness-dependent remanent polarization with respect to film thickness. (*Physical Review B* 84, 054112 (2011))

Comment 6. *Usually the mechanical properties of inorganic membranes are strongly related to the thickness, e.g. strain or elasticity. With this simple preparation method of PbZrO₃ membrane, the authors are suggested to prepare the AFE membranes with various thicknesses and compare their elasticity dependence on membrane thickness.*

Reply: Thanks for the comment. In the previous manuscript, we reported the super-elasticity of a 120 nm freestanding PbZrO₃ membrane. Here, following the suggestions by the reviewer, we have examined the flexibility of freestanding PbZrO₃ membranes with different film thicknesses. Thinner (~30 nm) and thicker (~133 nm) freestanding membranes were prepared, as the depth profiles shown in Fig. R7 (also newly added Fig. S12). Similarly, their flexibility is examined by the *in situ* SEM bending test.

For the thinner samples, the cut nanoribbons fold spontaneously due to the release of epitaxial strain, as shown in Fig. R7a (also newly added Fig. S12a). We pushed its one end the nanoribbon to bend in a small radius of curvature. The smallest bending radius is about 0.67 μm , corresponding to a bending strain of about 2.25%. It is difficult to achieve a 180° bending formation due to its original folding state. However, these bending tests have already manifested its high flexibility and good recoverability (see Fig. R8, also newly added Fig. S13). For the thicker samples, their nanoribbon could reach a nearly 180° folded state. The smallest bending radius is

about 1.86 μm , corresponding to a bending strain of about 3.57% (Fig. R9, also newly added Fig. S14). Thus, we could confirm the existence of this novel super-elasticity in our freestanding PbZrO_3 membrane. As the thickness becomes even larger, the density of flaws would increase in samples. Then the shape recovery ability could be significantly weakened because the cracking or dislocations are prone to initiate to induce a brittle fracture.

Fig. R7 (also newly added Fig. S12) **a** The SEM images of freestanding PbZrO_3 nanoribbon with a thickness of 30 nm. **b** The atomic force microscopy (AFM) depth profile of thinner (~30 nm, the left) and thicker (~133 nm, the right) freestanding PbZrO_3 membrane transferred on a silicon substrate. The inset shows the surface morphology.

Fig. R8 (also newly added Fig. S13) *In situ* SEM bending test of freestanding PbZrO_3 nanoribbon with a thickness of 29 nm. **a-b** The first to fourth columns respectively correspond to the initial, intermediate, maximum and residual bending states during four subsequent bending cycles. The last column shows the states after the removal of the external load. Scale bars, 5 μm .

Fig. R9 (also newly added Fig. S14) *In situ* SEM bending test of freestanding PbZrO₃ nanoribbon with a thickness of 133 nm. a-b The first to fourth columns respectively correspond to the initial, intermediate, maximum and residual bending states during four subsequent bending cycles. Scale bars, 5 μm.

Reviewer #2:

In this paper, authors have successfully fabricated a freestanding single-crystalline PbZrO₃ membrane, which is an antiferroelectric oxide. The freestanding membrane shows ultrahigh flexibility. Moreover, the elasticity and antiferroelectricity of freestanding PbZrO₃ membrane are investigated. The experimental results indicate that the freestanding PbZrO₃ membrane can accommodate a large bending strain of up to 3% without any crack or damage. Meanwhile, excellent dielectric and ferroelectric properties of freestanding PbZrO₃ membranes have been observed. This work is interesting, and I would recommend the article for publication after the following points are revised or explained:

We thank the reviewer for supporting its publication in this journal. We have presented a point-to-point reply to the comments. We also revised our manuscript and supplementary information accordingly.

Comment 1: *For Fig. 1g, the author described that “The corresponding selected-area electron diffraction (SAED) pattern shows typical 1/4 superlattice reflections...”. However, though the diffraction spots have been marked by arrows, it is hard to observe the “1/4 superlattice reflections” in the current figure. Maybe, the brightness and contrast could be readjusted to make the SAED pattern more clearly.*

Reply: Thanks for the suggestion. We have updated the SAED pattern in Fig. 1h to illustrate the 1/4 superlattice reflections more clearly (Fig. R10a, also updated Fig. 1h). As shown in Fig. R10b, similar results have been reported before.

Fig. R10 Corresponding selected-area electron diffraction pattern of freestanding PbZrO₃ membrane: **a** (also updated Fig. 1g) updated SAED patterns, **b** previous SAED patterns with adjusted contrast.

Comment 2: *In Fig. 2a, the curves of field-dependent dielectric permittivity of as-grown and freestanding thin films are significantly different. What is the reason for this difference? The release of strain from the substrate or other reasons? The*

authors are suggested to take more explanation for this phenomenon. Meanwhile, the author could add some description for four dielectric peaks to make the AFE-FE or FE-AFE transition process more clearly.

Reply: Thanks for the good comment. The dielectric permittivity of freestanding PbZrO₃ membranes increased obviously as compared to as-grown ones. Such an increase results from the release of lattice strain by the removal of the substrate clamping effect. Enhancement of dielectric permittivity after reducing or fully eliminating such clamping effect is widely observed in ferroelectric films (*Phys. Rev. Lett.* 108, 157604 (2012)), as shown in Fig. R11. Similar to those previous reports, as-grown PbZrO₃ thin films are also subjected to in-plane compressive strain since the lattice constant of PbZrO₃ crystals ($a_t = b_t = 4.161 \text{ \AA}$, and $c_t = 4.110 \text{ \AA}$, “t” denotes pseudo-tetragonal indices) is much larger than the SrTiO₃ substrate ($a = 3.905 \text{ \AA}$) as well as the sacrificial layer. Such lattice strain constrains the displacement of dipoles. The release of lattice strain (about 0.23%) is obvious when they become freestanding, as observed by X-ray diffraction patterns (Fig. 1d).

We observed four peaks in electric field-dependent dielectric permittivity curves. They all correspond to AFE-FE transitions, similar to classic anti-ferroelectric materials. The PbZrO₃ membranes/films are anti-ferroelectric at zero field. When a positive electric field was applied, the first peak emerged at about 291 kV/cm, corresponding to a typical AFE → FE transition. PbZrO₃ transforms to the ferroelectric state at the higher electric field. Another peak starts to appear at about 171 kV/cm removal of the applied electric field, corresponding to FE → AFE transition. Similar behavior could be observed when applying/removing a negative electric field. We have updated the ϵ - E curves in Fig. 2a to illustrate the scanning sequence more clearly, as mentioned by Reviewer #1. We also have added more discussion in the revised manuscript.

[Redacted]

Fig. R11 Enhancement of dielectric permittivity after removal of substrate clamping effect. (*Phys. Rev. Lett.* 108, 157604 (2012))

Comment 3: *As we all know, the inorganic ferroelectric thin films could not accommodate too large strain, typically below 2%. In this work, the freestanding thin film has a super elasticity, however the critical strain is about 3%. In the simulation section, the calculated strain is up to 9%, which is much larger than the critical strain of freestanding thin film. I think, it is necessary to explain the relationship between experimental and simulated results and the reason why such a large strain is adopted*

in the simulations. Moreover, in Fig. 4c, the vortex-like domain structure is observed. How to understand this result? Is it useful for explaining the experimental results?

Reply: As mentioned by the referee, the maximum recoverable strain could reach 9% for the PbZrO_3 membrane in atomistic simulations, much larger than that of ~3% observed in experiments. The main reason lies in the distinct difference of sample thickness between simulation and experiment. Owing to the high computing cost, we only create a thinner film with a thickness of 8 nm. However, the sample for the bending test has a thickness of 120 nm in our experiment. As the sample size reduces to the nano-scale, the size effect could become more dominant in affecting the shape recovery ability of materials. For example, San Juan et al. compared the superelasticity of the bulk Cu-Al-Ni shape-memory alloy and its micro-pillar counterpart (*Nat. Nanotechnol.* 4, 415-419 (2009)). They found that the micro-pillar has an ultra-high damping with a larger recoverable strain. Moreover, even the structural ceramics, which behave to be brittle in bulk, were discovered to exhibit novel shape memory and superelastic behavior at a small scale. For example, A. Lai observed that a zirconia pillar with a diameter of micrometer can withstand strains over 7%, far beyond the fracture strain in bulk samples. (*Science* 341,1505-1508(2013)) Therefore, we can see that as the sample size becomes smaller, the shape recovery ability is enhanced dramatically. This could lead to a higher recoverable strain for the small size in our atomistic simulation in comparison to that of experimental observations.

As indicated by the referee, there exist vortex-like structure domains in the bent PbZrO_3 membrane, as marked by the clockwise and anticlockwise circles in the revised Fig. R12. The formation of this vortex-like structure arises from the gradient strain under bending deformation. We know the sample experiences a transition from compressive to tensile strain near the neutral layer upon bending. The dipoles are then driven to rotate towards the directions normal to the neutral layer. As a result, a fully closed vortex-like domain structure is generated near the neutral layer. We noticed that a similar vortex structure is also observed in another system under bending deformation. Here, we presented unpublished simulation results of the formation of a similar vortex domain in a PbTiO_3 membrane (see Fig. R13). The vortex-like domains could take an important role in assisting the bending-induced super-elasticity. As we demonstrated in our previous work (*Science* 366, 475-479 (2019)), the continuous dipole rotations could help to eliminate the mismatch stress largely in the coexisting differently oriented domains at high strain, avoiding mechanical failure by the sharp domain switching.

We have added the relevant discussion in the revised manuscript.

Fig. R12 Domain evolution of freestanding single-crystalline PbZrO_3 membrane under bending. **a** Atomic configuration of the sample under the uniaxial tensile and compressive strains. The colors are coded according to the dipole direction. **b** The typical configuration at the local site, as marked in (a).

Fig. R13 Vortex domain structure formed in PbTiO_3 membrane under bending. The local vortex structures are identified in local regimes.

Reviewer #3:

The topic of freestanding antiferroelectric PZO is fascinating. The author used a few methods to explore the structure, functionality, and deformability of the PZO membranes. However, the results presented by the authors are insufficient to substantiate the conclusions they have reached. The 3% elastic strain the authors report is not remarkably distinct from that in other perovskite oxides. While the authors suggest that the transition from antiferroelectric to ferroelectric states contributes to elastic deformation, they fail to offer concrete experimental evidence. Additionally, the authors need to address the following questions:

Reply: Thanks for your careful reading. Your insightful comments and advice have helped us improve the manuscript adequately.

We believe a $>3\%$ bending strain in freestanding PbZrO_3 membrane is super-elastic for the following reasons. (1) Usually, the super-elasticity or other extraordinary mechanical behaviors of freestanding single-crystalline oxide membranes are compared with bulk materials. Generally, bulk ferroelectric and antiferroelectric materials are brittle and crack easily during bending, and this is widely studied in bulk ferroelectrics. They (either ceramics or single-crystals) can only endure a very small bending strain $<1\%$, as summarized in Fig. R14, and most of them are damaged when bending strain $>0.4\%$. (2) The poly-crystalline ferroelectric thin films are also brittle as they have too much imperfections such as domain boundaries. A recent study reveals that BaTiO_3 polycrystalline thin films deposited on Ni substrates initiate cracks when tensile strain $\geq 0.8\%$, as summarized in Fig. R15. The critical fracture strain for 300 nm, 600 nm and 900 nm BaTiO_3 films are 0.8%, 0.8% and 0.3%, respectively. (3) We notice that a $>3\%$ strain is even considerable to that realized in some metallic nanowires which have improved mechanical properties compared to their bulk, as summarized in Table R1.

In addition, the mechanically induced AFE-FE transition is well-studied in single-crystalline PbZrO_3 -based thin films both experimentally and theoretically. Chaudhuri *et al.* (*Physical Review B* 84, 054112 (2011)) and Gao *et al.* (*Appl. Phys. Lett.* 115, 072901 (2019)) found that the interfacial compressive strain/stress will stabilize a ferroelectric phase in PbZrO_3 -based epitaxial thin films on SrTiO_3 substrates, as directly observed by TEM (Fig. R16a-c). Correspondingly, double hysteresis loops turn to ferroelectric-like loops (Fig. R16d) with an increase of P_r (Fig. R16e). This is more evident in ultrathin PbZrO_3 films which experience larger compressive strain/stress, while such lattice strain would be released dramatically in thicker films. Theoretically, it is an elastic energy-driven AFE-FE phase transition (*ACS Appl Mater Interfaces* 14, 25770-25780 (2022)). Although we did not directly observe the microstructure evolution of such phase transition during bending freestanding PbZrO_3 , the P - E hysteresis loops show a small but non-negligible remanent polarization and it is consistent with those previous observations. We believe the intrinsic nature of the strain-induced AFE-FE phase transition in our freestanding PbZrO_3 membranes is the same as that of the strained PbZrO_3 thin films.

[Redacted]

Fig. R14 Representative stress-strain curves for bulk ferroelectric materials during bending tests. **a** Lead-based piezoelectric ceramics or single-crystals (Bending Strength of Piezoelectric Ceramics and Single Crystals for Multifunctional Load-Bearing Applications. *IEEE Transactions on Ultrasonics Ferroelectrics and Frequency Control* 59(6): 1085-1092 (2012)). **b** Lead-free piezoelectric ceramics (Deformation and bending strength of high-performance lead-free piezoceramics. *Journal of the American Ceramic Society* 105(5): 3128-3132 (2022)).

[Redacted]

Fig. R15 Damage tolerance of BaTiO₃ polycrystalline thin films. **a** Geometry and dimensions of the samples for test and the sample is Ni(substrate)/BaTiO₃(thin films). **b-c** surface SEM images of cracked films with different thicknesses (b) 300 nm BaTiO₃ and (c) 900 nm BaTiO₃. (Multiscale characterization of damage tolerance in barium titanate thin films. *Journal of Applied Physics* 132: 045302 (2022)) The critical fracture strains for 300 nm, 600 nm and 900 nm films are 0.8%, 0.8% and 0.3%, respectively.

Table R1 Summary of elastic strains and strengths of metallic nanowires. (The Mechanical Properties of Nanowires. *Advanced Science* 4: 1600332 (2017))

Nanowire materials	Maximum elastic strain	Testing method
Au	10%	AFM bending test
Au	19.6%	in situ TEM tensile test
Cu	5%	in situ TEM tensile test
Cu	7.2%	in situ SEM tensile test
Ag	4%	in situ SEM tensile test
Co	2.14%	in situ SEM tensile test

[Redacted]

Fig. R16 Lattice strain induced AFE-FE phase transition. **a** Cross-sectional TEM image of the $\text{Pb}_{0.97}\text{La}_{0.02}\text{Zr}_{0.95}\text{Ti}_{0.05}\text{O}_3$ (PLZT) epitaxial thin film. **b** SAED patterns at various depths and the selected areas are labeled in (a). (*Appl. Phys. Lett.* 115, 072901 (2019)) **c** Cross-sectional TEM image of a $\text{SrTiO}_3/\text{SrRuO}_3/\text{PbZrO}_3$ heterostructure with $d_{\text{PZO}} = 22$ nm and the insets show SAED patterns, indicating the orthorhombic phase (AFE phase) and the rhombohedral phase (FE phase). **d** thickness-dependent (left) dynamic and (right) static P - E hysteresis loops of PbZrO_3 epitaxial thin films with different thicknesses. **e** Thickness-dependent remanent polarization. (*Physical Review B* 84, 054112 (2011))

Comment 1: *On page 3, the authors describe, “Fig. 1b shows a millimetre-scale (2.5 mm × 2.5 mm) crack-free freestanding PbZrO_3 membrane transferred to a polydimethylsiloxane (PDMS) substrate, which still has high flexibility”. To substantiate their claim of no cracking, an SEM image or a high-magnification optical microscopy image is necessary.*

Reply: Thanks for the suggestion. We have collected the scanning electron microscope (SEM) images of freestanding PbZrO_3 membrane with different magnifications, as shown in Fig. R17 (also newly added Fig. S1). Clearly, the freestanding membranes remain integral and crack-free. We have provided these images in the revised Supplementary Information.

Fig. R17 (also newly added Fig. S1) SEM images of transferred freestanding single-crystalline PbZrO_3 membrane.

Comment 2: *Regarding perovskite oxides, there are a few studies documenting their exceptional deformability, even in single crystals. Similar behaviours have been reported in membranes, as referenced by the authors. Given that this paper focuses on outstanding deformation ability, it should include more in-depth discussions about the mechanisms of deformation in perovskite oxides.*

Reply: Thanks for the comment. This novel super-elastic behavior has been discovered in a few freestanding single-crystalline perovskite oxide membranes, such as BaTiO₃ (a classic FE material), BiFeO₃ (a typical multiferroic material) and here PbZrO₃ (a classic AFE material). Except for strain-induced AFE-FE transition in freestanding PbZrO₃, we think that the dipole switching especially the continuous dipole rotation upon a strain gradient, plays a critical role in the shape recovery of those deformed perovskite FE or AFE oxide. In freestanding BaTiO₃, large bending strain could induce continuous rotation of dipoles (Fig. R18), making the sample transform from a single-domain state to a multi-domain state. A transition zone with continuous dipole rotation forms near the neutral plane under bending. This continuous transition zone could largely eliminate the mismatch stress in the coexisting *c* and *a* nanodomains at high strain level, avoiding mechanical failure by the sharp domain switching. An extremely large bending strain could even lower the energy barrier between *a* and *c* domains pronouncedly, generating metastable states in which dipoles are able to stay with some deviations from *a* or *c* domains. For the super-elasticity of freestanding BiFeO₃ membranes, it originates from the rhombohedral-tetragonal phase transition, together with a rather thicker transitional zone where polarization rotates continuously between the thickness direction and diagonal direction of a pseudo-cubic unit cell. The continuous rotation could largely eliminate the mismatch stress caused by the abrupt change of microstructures at high strain level, like the AFE-FE transition here, avoiding the possible mechanical failure.

We have rewritten the whole discussion.

[Redacted]

Fig. R18 Continuous dipole rotation in freestanding BaTiO₃. **a** Experimental observation. **b** Theoretical calculation. (*Science* 366, 475-479 (2019)). In bulk tetragonal BaTiO₃, only *a* and *c* domains (horizontally/vertically aligned relative to film normal) exist. In freestanding BaTiO₃ dipoles rotate and align to other orientations.

Comment 3: On page 3, the authors state, “The atomic force microscopy (AFM) image reveals that freestanding PbZrO₃ membranes have a very smooth surface with a surface roughness of 2.26 nm (Fig. 1c).” However, based on the scale bar, it is evident that the surface roughness exceeds 2.26 nm.

Reply: The atomic force microscope (AFM) images are measured by Bruker Icon and the surface roughness is obtained from its software (NanoScope Analysis, version 1.8). We have double-checked the surface roughness again and confirmed it has a close value (Fig. R19).

Fig. R19 Surface morphology of transferred freestanding single-crystalline PbZrO₃ membrane by AFM.

Comment 4: Figure 1(D) on Page 4: The indexing of the XRD peaks appears to be incorrect, particularly for PZO, where the out-of-plane lattice parameter is double that of the cubic perovskite structure. Moreover, there is a shoulder peak for the PZO, which is likely attributable to the orthorhombic (240).

Reply: We have updated the indexing of the XRD peaks in Fig. 1d (also Fig. R20). The PbZrO₃ film presents a typical orthorhombic perovskite crystal structure with two diffraction peaks at (240)_o and (004)_o (“o” denotes orthorhombic indices), consistent with that reported in other literatures.

Fig. R20 (also updated Fig. 1d) X-ray diffraction patterns of as-grown SrTiO₃/Sr₃Al₂O₆/PbZrO₃ heterostructure and freestanding PbZrO₃ membranes on the platinized silicon substrate. a.u., arbitrary units.

Comment 5: *Figures 1(F, G) on page 4: The antiparallel displacement of Pb atoms is not visible in the HAADF-STEM image (panel F). Additionally, the 1/4 superlattice reflections in the FFT image (panel G) are much weaker than those reported in other studies of PZO.*

Reply: Thanks for the critical comment. We have collected more HAADF-STEM and selected area electron diffraction patterns. Still, the antiparallel displacement of Pb atoms is not obvious in the current HAADF-STEM images. Nevertheless, we can see clearly the 1/4 super-lattice reflections (Fig. R21), as we replied to Reviewer #2. After a thorough literature review, we found that the intensity of such super-lattice reflection is considerable to previous reports (Fig. R22).

We also obtained clues about the microstructure of freestanding PbZrO₃ from SAED patterns (*Appl. Phys. Lett.* 115, 072901 (2019)). The very weak $1/2\{110\}c$ and the obvious $1/4\{110\}c$ super-lattice reflection indicates the commensurate AFE phases with dipole aligns like $\uparrow\downarrow\uparrow\downarrow$ and $\uparrow\uparrow\downarrow\downarrow$, respectively. The notable elongation of $1/4\{110\}c$ along $[01-1]$ directions as $1/n\{110\}$ implies the incommensurate AFE phases with dipole arrangement like $\uparrow\uparrow\downarrow\downarrow\uparrow\uparrow\downarrow\downarrow$. From SAED patterns, we could observe an incommensurate periodicity of $n = 3.38\sim 4.76$ along the $[01-1]$ direction (with a corresponding modulation wavelength of 1.05~1.48 nm). The $1/4\{110\}c$ and $1/2\{110\}c$ commensurate AFE phases produce no remnant polarization (P_r), while the $1/n\{110\}c$ incommensurate AFE phases result in a non-zero P_r . Here, the directly measured P - E loops show a very small but non-negligible P_r , which confirms the coexistence of commensurate and incommensurate AFE phases in both as-grown heterostructure and freestanding PbZrO₃ membranes. Similar behavior has been widely observed in PbZrO₃-based antiferroelectric ceramics and films (*Materials Horizons* 7, 1912-1918 (2020)) (*Nat. Commun.* 12, 4215 (2021)). Since there is a mixture of commensurate and incommensurate AFE phases, the dipole alignment is very complex which makes it difficult to prepare suitable TEM samples. The shading effect may disturb a clear observation of the anti-parallel displacement of Pb atoms.

Currently, we are looking for collaborations with TEM experts to further clarify this issue.

Fig. R21 Corresponding selected-area electron diffraction pattern of freestanding PbZrO_3 membrane: **a** (also updated Fig. 1h) updated SAED patterns, **b** previous SAED patterns with adjusted contrast.

[Redacted]

Fig. R22 Lattice strain induced AFE-FE phase transition. **a** Cross-sectional TEM image of the $\text{Pb}_{0.97}\text{La}_{0.02}\text{Zr}_{0.95}\text{Ti}_{0.05}\text{O}_3$ (PLZT) epitaxial thin film **b** SAED patterns at various depths and the selected areas are labeled in (a). (*Appl. Phys. Lett.* 115, 072901 (2019)) **c** Cross-sectional TEM image of a $\text{SrTiO}_3/\text{SrRuO}_3/\text{PbZrO}_3$ heterostructure with $d_{\text{PZO}} = 22$ nm and the insets show SAED patterns. (*Physical Review B* 84, 054112 (2011))

Comment 6: *The structure of both the as-grown PZO thin film and the free-standing PZO membrane requires more thorough examination. The PZO structure is expected to be complex. Even in the PZO single crystal, a ferrielectric phase may appear.*

Reply: We agree that the structure of PbZrO_3 is really complex as previously reported, which could be affected by many factors like epitaxial strain, film thickness and chemical doping. Both the AFE phase and FE phase have been observed before in PbZrO_3 -based antiferroelectric materials. For our as-grown PbZrO_3 thin film and the freestanding PbZrO_3 membrane, we can confirm that: (1) They all have an orthorhombic crystalline structure. This is confirmed by both X-ray diffraction (Fig. 1d) and electron diffraction in TEM images (Fig. 1h). (2) Both commensurate and incommensurate AFE phases coexist as we have discussed above in response to the last comment.

As we have addressed in response to your comment #2, fundamentally, dipole switching especially the continuous dipole rotation upon a strain gradient, plays a critical role in enabling such super-elasticity in perovskite AFE/FE oxide. Continuous dipole rotation in AFE PbZrO_3 could induce other unique dipole configurations. It was recently reported that both the magnitude and the relative angle between neighboring 180° FE domains may be different and could be accommodated by ferrielectric (FiE) phase. The FiE phase could behave in either the magnitude modulation mode or the angle modulation mode. In our simulations, we found both modes exist at local sites. Bending deformation may promote the formation of such FiE phase with uncompensated polarization as observed experimentally (Fig. 2e).

We have re-written the discussion after your comments.

Comment 7: *The hysteresis loops depicted in Figure 2(B) appear significantly more complex than typical antiferroelectric loops. The authors should provide an explanation for this occurrence.*

Reply: Thanks for the suggestion. We have observed typical double hysteresis loops for antiferroelectric materials in previous measurement (Fig. R23a), but they seem not very smooth. We now carried out more hysteresis loop tests and obtained much smoother ones (Fig. R23b, also updated Fig. 2b). Besides, we have applied an even higher electric field ($>1\text{MV/cm}$) as suggested by Reviewer #1 (Fig. R3 (also newly added Fig. S5)). Such double hysteresis loops, corresponding switching current curve (Fig. R23c) and four peaks in electric field-dependent dielectric permittivity curves (ϵ - E curves, see updated Fig 2a) present the typical characteristic of AFE materials, similar to previous reports for PbZrO_3 -based thin films in literatures (Fig. R24). The hysteresis loops have been updated in the revised manuscript (Fig. 2b).

Fig. R23 a-b Electric field dependent P - E hysteresis loops of as-grown and freestanding PbZrO_3 membranes in previous measurement (a) and much smoother

ones (b) (also updated Fig. 2b). c The switching current curve of as-grown and freestanding PbZrO₃ membranes.

[Redacted]

Fig. R24 Typical double *P-E* hysteresis loops for undoped PbZrO₃ films. a SrTiO₃(001)/LSMO/PbZrO₃(50 nm). (*Nat. Commun.* 12, 4215 (2021)) **b** SrTiO₃(001)/LSMO/PbZrO₃(50 nm). (*Appl. Phys. Rev.* 10, 011403 (2023)) **c** SrTiO₃(001)/LSMO/PbZrO₃(110 nm). (*ACS Appl Mater Interfaces* 14, 51096-51104 (2022))

Comment 8: *Figure 2(D): The colors of the different loops appear quite similar. It is suggested to use colors that are distinctly different for better clarity.*

Reply: We thank the reviewer for the good suggestion. We have updated the colors of *P-E* loops in Fig. 2d (also Fig. R25) to make them more clearly. We also provide separate loops in Fig. S8 (also Fig. R26) in Supplementary Information. Furthermore, we have examined the *P-E* hysteresis loops of the freestanding PbZrO₃ membranes with larger bending strain. We transferred the PbZrO₃ membranes on the indium tin oxide coated PET substrate which is thicker than the platinized PI substrate. We then attached them to molds with different radii as 10 mm, 5 mm and 2.5 mm, corresponding maximum strains are about 0.83%, 1.66% and 3.3%, respectively (see updated Fig. S7).

Fig. R25 (also updated Fig. 2d) The *P-E* hysteresis loops of the freestanding PbZrO₃ membranes on the ITO-coated PET substrate during bending.

Fig. R26 (also newly added Fig. S8) The *P-E* hysteresis loops of the freestanding PbZrO₃ membranes on the platinized PI substrate during bending.

Comment 9: *More details should be provided regarding the bending deformation (Fig. 2(C)), such as the geometry of the mold, how to put the sample on it, how to bend it, etc.*

Reply: We have provided the optical images of the molds we used (Fig. R27, also newly added Fig. S6) and the whole test fixture in the Supplementary Information. We also provided more details in Methods. A ferroelectric analyzer (aixACCT TF2000) was employed to evaluate the antiferroelectric properties. After transferring the freestanding PbZrO_3 membrane on the flexible ITO-coated PET substrate, the negative photoresist (SU8) was spun-coated on both sides of it as a fixing layer (3000 rpm, 50 s). We then attached it to molds with different radii using tape or double-sided tape to test its antiferroelectric properties.

Fig. R27 (also newly added Fig. S6) Bending test molds with different radii as 10 mm, 5 mm and 2.5 mm.

Comment 10: *Ferroelastic domain switching, which was reported in ferroelectric materials to contribute to elastic deformation, should be considered.*

Reply: Thanks for your valuable suggestion. As we mentioned above, in freestanding PbZrO_3 membrane, both AFE-FE transition and continuous dipole rotation are responsible for super-elasticity. Especially, continuous dipole rotation will induce small non- 180° FE domains by ferroelastic domain switching and we indeed observed the ferroelastic domain switching in our simulations. In order to present this point clearly, we have calculated the atomic-level strain. Fig. R28 (also newly added Fig. S19) shows the map of normal strain ϵ_{xx} upon mechanical deformation. With the initial stress-free configuration as a reference state, we found ferroelastic domains laying inside the plane are generated in the tensile zones (red patches), while those laying outside of the plane are generated in the compressive zones (blue patches). The ferroelastic switching, coupled with the electric dipole, could contribute to the good mechanical properties of our freestanding PbZrO_3 membrane.

Fig. R28 The strain map for the freestanding single-crystalline PbZrO_3 membrane under mechanical deformation. **a** Uniaxial tension and compression. **b** (also newly added Fig. S19) Bending. The color represents the normal strain ϵ_{xx} along the x direction.

Comment 11: *In the simulation, it appears the authors did not take into account the impact of ferroelastic switching and intermediate ferrielectric phases. Without experimental evidence, it's uncertain whether these results accurately represent actual conditions.*

Reply: Thanks for the comment. Several previous research confirms the ferrielectric (FiE) phase when the AFE order parameter is coupled with polarization (*Nature Commun.* 11, 3809 (2020) *Nano Letters* 23(4): 1522-1529 (2023); *Advanced Materials* 35(3): 2206541 (2023)). As we replied to the last comment, the ferroelastic domain switching does exist in our simulations. In addition, the intermediate ferrielectric phases can be observed in the transition regime between compressive ferroelastic domains (blue color) and tensile ferroelastic domains (red color). We have added the relevant discussion in the revised manuscript.

In addition, we also confirmed the super-elastic deformation behavior in freestanding PbZrO_3 with different film thicknesses (Fig. R29). We observed an even larger maximum bending strain $\sim 3.57\%$. We also believe continuous dipole rotation could form such FiE phase and contribute to uncompensated polarization as observed experimentally (Fig. 2e). It is highly possible that such an intermediate phase may exist in the transition zone which needs further careful verification. We are trying to prepare suitable TEM samples with a large fixed bending strain. However, fixing a large bending strain in freestanding membranes is difficult due to their good shape recoverability (Figs. 3a-3c). We are looking for collaborations to characterize such intermediate states experimentally by TEM.

Fig. R29 (also newly added Fig. S14) *In situ* SEM bending test of freestanding PbZrO_3 nanoribbon with a thickness of 133 nm. a-b The first to fourth columns respectively correspond to the initial, intermediate, maximum and residual bending states during four subsequent bending cycles. Scale bars, 5 μm.

REVIEWER COMMENTS

Reviewer #1 (Remarks to the Author):

My questions have been replied appropriately.

Reviewer #2 (Remarks to the Author):

The authors have addressed all my comments so I recommend the revised manuscript to be accepted for publication.

Reviewer #3 (Remarks to the Author):

The authors have significantly enhanced their manuscript, yet I cannot agree with the statement of the super elasticity. Here are two papers on the deformation of ferroelectric materials: *Acta Materialia*, 181: 501 (2019); *Nature Communications*, 13: 335 (2022). Both papers report more substantial elastic deformation.

Response to reviewers

Reviewer #1: *My questions have been replied appropriately.*

Reviewer #2: *The authors have addressed all my comments so I recommend the revised manuscript to be accepted for publication.*

Reply: We thank all the reviewers for the careful evaluation and acceptance of our revised manuscript. This is a great encouragement.

Reviewer #3: *The authors have significantly enhanced their manuscript, yet I cannot agree with the statement of the super elasticity. Here are two papers on the deformation of ferroelectric materials: Acta Materialia, 181: 501 (2019); Nature Communications, 13: 335 (2022). Both papers report more substantial elastic deformation.*

Reply:

Thanks for your critical comment and for providing more references to better interpret our results. We have carefully read those papers and confirmed that the elastic strains (~5% for BaTiO₃ and ~8% for PMN-PT) are indeed larger than that achieved in freestanding antiferroelectric PbZrO₃ membranes. The words "super" and "superior" sound not very objective and we have removed such expressions. In this study, we have achieved ~3.5% maximum bending strain in freestanding PbZrO₃ membranes. Since this deformation includes both elastic and plastic strain, we think it is more appropriate to use the term "flexibility" rather than "elasticity".

Moreover, your comment also reminds us that our initial goal is to demonstrate excellent flexibility and fundamental mechanisms in antiferroelectric oxide, as we have stated in the abstract and introduction before. Such a >3% bending strain in freestanding PbZrO₃ is reproducible (Fig. 3 and Fig. S14) and it is one order of magnitude larger than bulk ceramics, which is usually only 0.2%~0.4% (*IEEE Transactions on Ultrasonics Ferroelectrics and Frequency Control* 59(6): 1085-1092 (2012); *Journal of the American Ceramic Society* 105(5): 3128-3132 (2022)). Based upon the above observations, we believe that we have demonstrated a remarkable flexibility of antiferroelectric oxide.

Therefore, we have changed the title from "Superior Elasticity in Freestanding Single-crystalline Antiferroelectric PbZrO₃ Membranes" to "Remarkable Flexibility in Freestanding Single-crystalline Antiferroelectric PbZrO₃ Membranes" and modified relevant description in the main text. Thanks again for all your insightful comments.

REVIEWERS' COMMENTS

Reviewer #3 (Remarks to the Author):

The authors have adequately addressed my concerns. I have no remaining questions and suggest its publication in Nature Communications.